# Exploring gender-intentional implementation of Digital Health Information for Immunization in Ethiopia

Getasew Amare[1], Betelhem Abebe Andargie[1], Patricia Mechael[2,3]*, Xueli Qiu[3], Michelle R. Kaufman[3], Berhanu Fikadie Endehabtu[1], Kassahun Alemu[1], Erica Layer[2], Sarah Cunard Chaney[2], Binyam Tilahun[1]

**1** Center for Digital Health and Implementation Sciences, University of Gondar, Gondar, Ethiopia, **2** health.enabled, Washington, District of Columbia, United States of America, **3** Department of Health Behavior and Society, Bloomberg School of Public Health, Johns Hopkins University, Baltimore, Maryland, United States of America

\* patty@healthenabled.org

## Abstract

Despite growing investment in digital health for immunization, there is limited evidence on how gender is considered into design and implementation. In 2022, Ethiopia launched a Gender-intentional Digital Health Information Roadmap with the support of Gavi, the Vaccine Alliance; however, the operationalization of gender strategies remains unclear. This qualitative study explored gender inclusiveness in digital health for immunization in Ethiopia, drawing on document reviews, participatory workshops, key informant interviews, and stakeholder analysis across selected three regions in Ethiopia. Findings revealed that women dominate frontline roles as caregivers and vaccinators but remain underrepresented in supervisory, leadership, and digital system development positions. Although gender strategies and units exist, implementation is weak, with limited gender awareness and no intentional planning. In addition, a notable gender digital divide persists. Senior leadership role assignments, despite competence, depend on ability for women to balance responsibilities and travel with household duties. The routine health data systems, like District Health Information Software, version 2, lack sex-disaggregated immunization data, hindering gender-informed decision making. Enabling factors include national prioritization of gender equity, empowerment programs, and partner support, whereas barriers include cultural norms, inadequate advocacy, weak accountability, and insufficient resources. Embedding gender-transformative approaches is vital for equitable, inclusive digital health information and associated health system and immunization outcomes in Ethiopia.

**Data availability statement:** All relevant data necessary to replicate the results of the study are included in the Supporting information files.

**Funding:** This work was supported by Gavi, The Vaccine Alliance (Grant Number HSIS 12764 4 23) awarded to GA, BAA, PM, XQ, MK, BFE, KE, EL, SCC, and BL as part of the award to implement the Gavi Digital Health Information (DHI) Monitoring, Evaluation, and Learning (MEL) Plan. The funders had no role in study design, data collection and analysis, decision to publish, or preparation of the manuscript. No salaries were received in support of this work.

**Competing interests:** The authors have declared that no competing interests exist.

## Introduction

Digital health interventions have gained global recognition as effective tools for strengthening immunization systems through improved program planning, service delivery, and timely decision-making [1]. In addition, these interventions present opportunities to address structural gender inequalities and promote more equitable health outcomes by transforming underlying gender norms and barriers [2].

In 2022, the Ethiopian government, supported by Gavi, the Vaccine Alliance, developed a Gender-intentional Digital Health Information (DHI) for Immunization Roadmap. This roadmap outlines priority digital health interventions, digital health enablers, and gender-related activities to improve immunization coverage [3]. Since 2020, the Ethiopian Ministry of Health has issued strategic plans and guidelines to promote gender integration in immunization and digital health [3–9]. Similarly, other countries in Africa, including Kenya and Rwanda, have recognized the role of gender in digital health and immunization strategies [10,11]. However, the implementation of these gender-related activities remains limited, resulting in limited translation of policy commitments into practice.

The gender context in Ethiopia showcases a blend of advancements and persistent obstacles. Historically, the culture has been dominated by patriarchal norms, which tend to put women and girls at a disadvantage, particularly in areas like education, job opportunities, political engagement, healthcare access, and technology. Nonetheless, considerable strides have been made in recent years to foster gender equality [12]. Inequalities are also evident within the health and immunization workforce, where women constitute a significant portion of frontline healthcare providers but encounter barriers to leadership and decision-making roles.

Digital health has the potential to enhance immunization services, but its effectiveness is dependent on being designed, deployed, and implemented with a gender-intentional approach [13,14]. This is largely due to a significant gender digital divide that is manifested in mobile phone ownership, internet use, and digital literacy; in Ethiopia, the gender disparities in digital access remain pronounced, with a national average of only 52% of women owning a mobile phone compared to 79% of men and even wider gaps in rural areas [15,16]. Although the Ethiopian government has made considerable progress in emphasizing and advancing digital health initiatives, there remains a notable gap in understanding how to develop and implement these initiatives from a gender-intentional perspective [17].

Gender-intentional digital health approaches consider and address gender norms, roles, and power relations as critical for ensuring equitable access to and benefits from such technologies, particularly for women and marginalized groups. The ultimate goal of gender-intentional programming is to eliminate the systemic and harmful effects of gender norms, roles, dynamics, and relationships that cause gender inequities so that all users and beneficiaries have meaningful representation, engagement, and benefit from digital health interventions [18]. To begin to move incrementally along the continuum of gender transformative programming, gender sensitive programming includes interventions and activities that recognize gender norms and roles but do not take any specific action to address inequalities. Gender responsive

programming incorporates activities that seek to achieve gender-related programme goals and improve immediate out-puts. Gender transformative programming represents implementation of strategies and activities that address the causes of gender inequity to promote meaningful change [18].

Despite this, the gender context within Ethiopia's digital health system remains underexplored. Understanding gender and gender dynamics in digital health for immunization is crucial for assessing the sensitivity and responsiveness of these systems to the professional and societal context and needs of women and to develop more effective design and implementation strategies.

This study aimed to explore the gender balance among DHI for immunization stakeholders, gender dynamics, gender intentional data use and decision making, and barriers and enablers of gender-intentional DHI implementation within the immunization program.

## Materials and methods

### Ethics statement

Ethical approval for the study was obtained from the Institutional Review Board (IRB) of the University of Gondar (Ref: VP/RTT/05/84/2023), ensuring that the research adhered to ethical standards and guidelines.

### Study design

This study employed a descriptive interpretative qualitative design to explore the experiences, perspectives, and behaviors of key stakeholders involved in DHI for immunization. This approach enabled a detailed understanding of participants' lived experiences, capturing underlying themes, motivations, and barriers shaping their decisions. The study was informed by the Rapid Guide for Analysis, Planning, and Monitoring Guide for Gender-Intentional Digital Health Intervention & Enablers developed by Gavi and health.enabled [18]. This framework guided the study design and analysis of gender considerations and stakeholder engagements across various components of the digital health and immunization program in Ethiopia.

### Study settings

The study was conducted in three purposefully selected regions of Ethiopia: Addis Ababa City Administration, Amhara, and Sidama regions. These regions were chosen to represent diverse geographic and contextual settings, including urban and agrarian areas, as well as the variation in immunization service delivery performance (high and low), as identified in the 2023 National Comprehensive Immunization Program Evaluation. Selection was also guided by the availability of key digital health interventions including, District Health Information Software, version 2 (DHIS2), an open-source health management information system used for data collection, management, and analysis at various levels of Ethiopia's health system; Electronic immunization registry (EIR), a digital system designed to capture, store, and manage individual immunization records to support timely and complete vaccinations; Electronic Community Health Information System (eCHIS), digital tool used by Health Extension Workers in Ethiopia to record and report community-level health service data in real time; and mBrana, a mobile-based Logistics Management Information System used in Ethiopia to support vaccine and supply chain management, including stock tracking and distribution, as recommended by the Ethiopia DHI for Immunization Roadmap [3].

Within each region and city administration, one woreda and two health facilities were selected, along with relevant institutions at the national and regional level, including the Ministry of Health, Regional Health Bureaus, and Woreda Health Offices, one from each selected region and city administration.

### Data collection

Multiple qualitative data sources including document review, a participatory design workshop, key informant interviews, and stakeholder mapping were utilized to ensure data triangulation and strengthen the validity of the findings.

A structured document review was first conducted, analyzing national strategies, guidelines, policies, and manuals related to DHI for immunization through a gender lens and to assess degrees of gender intentionality (S1 Table). The review focused on identifying gender-sensitive language, representation, barriers, enablers, and inclusiveness.

The document review was followed by a one-day participatory workshop in Addis Ababa that brought together 21 experts in immunization, digital health, and gender from the Ministry of Health, Regional Health Bureaus, and development partners working in digital health and immunization. The workshop invited participants to share their perspectives and experiences related to gender and digital health in the immunization program through a series of small group activities, validated and strengthened the initial findings from the document review, and refined the research focus.

Stakeholder mapping was conducted to assess gender balance and representation across key groups involved in immunization and digital health. Using a structured checklist, this activity quantified and analyzed the current gender distribution among policymakers, digital health developers and implementers, gender experts, supervisors, and frontline vaccinators at different levels of the health system. This activity generated system-level data on gender representation and positioning within immunization, digital health, and gender-related structures.

We conducted key informant interviews with 30 participants across various levels of the health system and health workforce striving for an equal number of male and female respondents when possible. These included immunization program leads at the Ministry of Health and Regional Health Bureaus who were responsible for strategic planning, policy development, and resource allocation; digital health leads and experts for immunization including those involved in the implementation, management, and scaling of digital health solutions for immunization programs; gender experts who were responsible for leading gender-related initiatives within the health system at various health system levels; immunization program supervisors who oversee service delivery, monitoring quality, and addressing challenges; vaccinators who are directly administering vaccines at health facilities or during outreach sessions. Interview participants were selected purposely based on predefined roles, expertise, and relevance to the study objectives and on their direct involvement in the development, implementation, supervision, or use of digital health information systems for immunization or gender-related programming within the health sector. Recruitment was conducted through targeted invitations to individuals meeting these criteria. Interviews were conducted in person in the selected regions and health facilities using a face-to-face approach separately. Before each interview, the study's purpose was explained, and informed consent was obtained from the participants. On average, the interviews lasted 30–60 minutes depending on the roles and responsibilities of the key informants. A total of four trained data collectors (two male and two female) conducted the qualitative interviews in Amharic to ensure clarity and comfort for the participants. To foster open dialogue and minimize social desirability bias, we also made efforts to match interviewers and participants by gender.

## Data collection tools

A range of data collection tools were developed to support data extraction for the document review as well as to guide the key informant interviews. A document extraction checklist was used to assess the policy documents such as strategies, guidelines, procedures, directives, reports, checklists, and other relevant documents from their gender inclusive strategies perspective. Workshop minutes were also used to capture insight from the participatory workshop. In addition, a semi-structured interview guide was developed to explore participants' experiences and practices related to gender, the gender digital divide, gender representation, and policy responsiveness. It also examined barriers, enablers, and recommendations from their perspectives. A checklist was used in the stakeholder mapping to assess gender representation among various stakeholders across different sectors, roles, and expertise.

## Data analysis

All interviews were conducted in Amharic, audio-recorded with consent, and labeled systematically. Transcriptions and translations into English were performed by the interviewers to ensure contextual accuracy. The translated transcripts

were then double-checked and revised by a quality assurance expert to maintain accuracy and reliability. An inductive thematic analysis was employed using Atlas.ti software. A codebook was developed using 20% of transcripts, selected to reflect diversity in respondents and regions. Three coders independently reviewed these transcripts and generated initial code lists. Inter-coder agreement was assessed using Krippendorff's alpha, yielding a score of 0.732, which indicates substantial agreement. Codes were then clustered into sub themes and synthesized into thematic categories aligned with the study objectives. Findings form the interviews were triangulated with the results from the document review, workshop reports, and stakeholder mapping to enhance analytical depth and ensure consistency.

## Results

### Study characteristics

A total of eight key documents on immunization and digital health from 2020 to 2023 were reviewed for this study, all issued by the Ethiopian Ministry of Health. The documents varied in type, including strategies, guidelines, and manuals, and applied a gender lens to different extents. While several explicitly identified gender disparities such as limited access to health services for women, low digital literacy, and under representation in leadership, others addressed equity in general terms. The *Ethiopia Strategy for Mainstreaming Gender within Digital Health and Health Information System* and the *Ethiopia National Gender Mainstreaming Manual for Health* offered the most comprehensive strategies including institutionalized gender budgeting, and equitable work participation (S1 Table). The desk review of national policies and strategies also showed a gender gap in representation in relevant policy development processes. It was identified that immunization and digital health strategic documents were predominantly developed by men. Specifically, out of the 70 authors involved for development of the *Ethiopia National Expanded Program on Immunization Comprehensive Multi-year Plan, more than 55 were men.* While the *Routine Immunization Catch-up Vaccination Guidelines* were developed by 22 experts, all of whom were men. The review also highlighted a persistent gender digital divide: only 57% of women owned a mobile phone compared to 79% of men. A total of 21 stakeholders also participated in the participatory workshop to validate findings from the desk review, comprising 12 women and 9 men from a range of national and international institutions.

A total of 30 key informants, 18 females and 12 male, were also interviewed including policy makers, digital health experts, gender experts, supervisors, and vaccinators from the three selected regions. Policy makers were predominantly males, while most digital health and gender experts were females. Most of the participants held advanced degrees (MPH, MSc) and had extensive experience ranging from 6 to 22 years. Immunization program supervisors were predominantly male possessing MPH degrees and 9–12 years of experience. In contrast, a majority of the included vaccinators were females aged 26–32, holding either BSc or diploma-level qualifications, with 5–9 years of experience (Table 1).

### Gender dynamics in Digital Health Information for Immunization

Findings from interviews, document reviews, and a stakeholder workshop revealed ongoing gender disparities in immunization and digital health programs, despite growing awareness of gender issues in healthcare. While strategic documents for immunization and digital health recognized these gaps and outline proposed solutions, implementation challenges remain. In addition, there are gender-specific departments established as dedicated units within the health system from the Ministry of Health to woreda health offices aiming to integrate gender considerations into program implementation and address inequalities. Women's empowerment initiatives including affirmative action and career development opportunities also exist within the broader health system. However, according to the interviews, this study found that the actual implementation of these initiatives within immunization and digital health programs is limited, both in terms of execution and measurable outcomes. The sections that follow provide deeper insights into these gaps, illustrating how gender gaps manifest across different levels of the DHI for immunization ecosystem.

**Table 1. Key-informant participant characteristics, 2024 Ethiopia.**

| Codes | Sex | Age | Educational status | Years of experience | Position/role in the DHI | Region |
|-------|-----|-----|-------------------|---------------------|--------------------------|--------|
| 014 | F | 27 | BSc | 5 years | Vaccinator | Amhara |
| 015 | F | 29 | BSc | 6 years | Vaccinator | Amhara |
| 016 | F | 32 | BSc | 9 years | Vaccinator | Amhara |
| 017 | F | 31 | BSc | 7 years | Vaccinator | Amhara |
| 018 | F | 30 | BSc | 6 years | Vaccinator | Sidama |
| 019 | F | 29 | BSc | 6 years | Vaccinator | Sidama |
| 020 | F | 27 | Diploma | 5 years | Vaccinator | Sidama |
| 021 | M | 29 | BSc | 6 years | Vaccinator | Addis Ababa |
| 022 | F | 30 | BSc | 7 years | Vaccinator | Addis Ababa |
| 023 | F | 26 | BSc | 5 years | Vaccinator | Addis Ababa |
| 024 | F | 36 | MPH | 11 years | Supervisor | Amhara |
| 025 | M | 37 | MPH | 12 years | Supervisor | Amhara |
| 026 | M | 36 | MPH | 10 years | Supervisor | Sidama |
| 027 | M | 36 | MPH | 9 years | Supervisor | Addis Ababa |
| 028 | M | 28 | BSc | 6 years | Supervisor | Addis Ababa |
| 029 | M | 42 | MPH | 19 years | policy maker | Amhara |
| 030 | M | 45 | MPH | 22 years | policy maker | Sidama |
| 031 | M | 38 | MPH | 13 years | policy maker | Addis Ababa |
| 032 | F | 41 | MPH | 18 years | policy maker | MoH |
| 033 | M | 43 | MSC | 20 years | policy maker | MoH |
| 034 | M | 36 | MPH | 12 years | Digital expert | Amhara |
| 035 | M | 37 | MPH | 11 years | Digital expert | Sidama |
| 036 | F | 29 | MPH | 7 years | Digital expert | Addis Ababa |
| 037 | F | 28 | MSC | 6 years | Digital expert | MoH |
| 038 | F | 39 | MSC | 14 years | Digital expert | MoH |
| 039 | F | 42 | DVM | 17 years | Gender expert | Amhara |
| 040 | F | 39 | MSC | 15 years | Gender expert | Sidama |
| 041 | F | 36 | MSC | 11 years | Gender expert | Addis Ababa |
| 042 | F | 38 | MSC | 15 years | Gender expert | MoH |
| 043 | M | 37 | MSC | 12 years | Gender expert | MoH |

**Gender understanding and gender-based role assignment in the workplace.** This study found that most participants, except for experts who were working in gender units, showed limited gender awareness across the health system. Many of them defined gender narrowly as the biological distinction between males and females, without considering the broader social, cultural, and structural factors that shape gender roles and experiences. One participant noted that:

> "I don't have detailed information about gender. My understanding of gender is simply that it refers to being female or male, and beyond that, I don't have much knowledge on the topic." (P021, vaccinator, M)

However, some participants, particularly those working in gender units, were able to provide a better explanation of the concept of gender. They discussed its various perspectives and emphasized its importance for the development of institutions and organizations, highlighting how gender awareness and integration can drive growth, inclusivity, and overall progress within these settings. The lack of training may contribute to the observed gap in gender awareness. The majority

of participants mentioned that they had little to no prior experience with discussions or training on gender topics. Because of this, their understanding of gender was basic and lacked a deeper understanding of its complexities and different viewpoints.

The study found a significant gender gap in how roles are assigned within immunization and digital health information. Tasks are generally allocated based on availability and qualifications, without acknowledging existing gender differences. The idea was explained by a participant as:

*"From my understanding, there is no specific principle that considers gender when assigning tasks; it's all based on competency and job requirements" (P028, supervisor, M)*

### Gender balance of stakeholders involved in Digital Health Information for Immunization

The gender balance among key stakeholders involved in digital health for immunization was analyzed using a stakeholder analysis matrix. Key stakeholders involved in DHI for Immunization included: vaccinators, supervisors, policy makers, digital health experts, and gender experts.

The findings revealed that women predominantly participate in caregiving and frontline health services, while men occupy most leadership, supervision, and digital health development positions. Over 95% of individuals responsible for bringing children for vaccination services are women. Similarly, more than 85% of community representatives who link caregivers with health services are female. Women also make up over 70% of vaccinators, indicating their significant role in direct immunization service delivery. However, women's representation in leadership and supervisory positions remains low, accounting for less than 35%. The disparity is even more pronounced in digital health development, where less than 1% of developers are women, and only about 30% are involved in leading digital health intervention implementation, despite women constituting more than 70% of end-users. Furthermore, women are underrepresented in immunization policymaking, holding fewer than 25% of decision-making roles**.**

Policymakers are aware of these gender gaps in digital health information for immunization and have taken several actions to address them; including affirmative action measures [19]. For example, female applicants receive an additional 5% bonus awarded to the total recruitment score. Respondents acknowledged that while this measure alone may not be a comprehensive solution, it represents a meaningful step toward achieving gender balance in the health workforce. A policymaker summarized this approach by stating:

*"As part of our affirmative action policy, we apply a 5% bonus to the scores of female applicants during staff recruitment to promote gender balance and ensure fair opportunities in roles where women are underrepresented." (P038, policy maker, F)*

Policymakers recognized the predominance of women in frontline service delivery roles, such as vaccinators and Health Extension Workers. However, they showed limited understanding of the structural and systemic barriers that contribute to women's underrepresentation in leadership, management, and decision-making roles in immunization and digital health. Instead, as a policymaker quoted here explained, some framed this gendered distribution of roles as deliberate and alignment with the needs of service users:

*"…when it comes to vaccinators, especially Health Extension Workers, almost all are women to better align with the female caregivers they serve in the country." (P030, policy maker, M)*

**Gender inclusiveness in Digital Health Information for Immunization.** Findings related to gender inclusiveness were organized by stakeholder group to reflect differences in experiences, awareness and practice. The following

sub-sections present perspectives from policy makers, digital health professional, supervisors, and vaccinators on the integration of gender with in the DHI for immunization.

**Policy-makers' perspective.** The assessment highlighted several gender-focused activities from the policymakers' perspective. Respondents' highlighted the Health Extension Program (HEP) design and prioritization of female vaccinators as key gender-responsive interventions, recognizing that most caregivers are mothers. Despite existing gender-focused initiatives in the health system, respondents indicated that the integration of gender considerations into the DHI for immunization program remains limited. A policy maker from the Ministry of Health unit stated:

*"Immunization activities are traditionally planned and implemented with little consideration for gender. Limited awareness and lack of guidance on gender-intentional planning are major barriers even I wasn't aware of these concepts until you explained them. (P043, policy maker, M)*

The findings also revealed gender disparities in various aspects of immunization program implementation. While strategic documents acknowledge gender disparities, they are primarily developed by men. Additionally, the gender gap is evident in leadership, as most policymakers and all EPI regional coordinators are male. Participants explained that policy-making roles require extra effort, time, and responsibility, making it harder for women to take on these positions due to their additional household and social responsibilities. As a result, many women are less likely to pursue or accept these leadership positions. A female policy maker from MoH shared her experience as:

*"As women, our household and social responsibilities take up a lot of time, making it hard to take on leadership roles. The sacrifices are big, and balancing everything is challenging. For example, I refused a director position because it would have been too difficult with my family duties" (P041, policy maker, F)*

**Digital health professionals' perspectives.** Participants indicated that gender inclusiveness in digital health for immunization remains limited, particularly in the participation and representation of women in DHI system development and implementation. The gender balance among digital health developers is highly skewed toward men with some participants noting that their offices lack female developers entirely. The participants added, with digital health interventions primarily designed and developed by men, female perspectives are often overlooked, potentially leading to systems that do not fully consider the unique challenges women may face in using or implementing these technologies. The participant described the idea as:

*"Most of the digital health systems are developed by men, and since females' engagement is limited, women's perspectives are often missing. This makes it hard to address the unique challenges women face when using or implementing these technologies." (P037, digital health professional, F)*

Additionally, the participants noted that because most frontline vaccinators are females, the digital health interventions at service delivery points tend to target them by default. However, they explained that the deployment, distribution, and capacity-building efforts for end users are carried out without integrating gender-specific considerations. One participant supports this idea as:

*"… during digital health allocation and capacity building activities we simply approach those available at the service delivery points whether they are male or females and we didn't have any specific approach considering gender" (P039, digital health professional, F)*

**Supervisors' perspectives.** The findings from the supervisors revealed that, although digital health tools are recognized as simple, easy to implement, and beneficial for the immunization program, the implementation process did not incorporate any gender-intentional approach.

Most participants acknowledged that digital health tools are essential for monitoring immunization services and managing supplies. However, they noted that capacity-building and supervision efforts do not incorporate gender-specific considerations. Respondents highlighted several gender-related challenges, including limited access to digital devices for female health workers, lower digital literacy levels among women, competing responsibilities like childcare affecting their participation in trainings, and safety concerns when traveling for supervision or data collection.

*"…there is a gap and difference in adopting these digital interventions. Women might take time to familiarize and quickly adopt the digital interventions because of the double burden she has at their house and work place. Thus, they tend to focus only on the manual paper based system, while digital technologies are very easier for men." (P024, supervisor, F)*

Additionally, deployment, distribution, and training processes are carried out without specific strategies to address these challenges. One supervisor stated that:

*"…we are implementing different digital health systems in the immunization program for program monitoring, but I didn't have any observation and experience in implementing them using gender intentional and focused approach." (P019, supervisor, F)*

Participants also emphasized that female vaccinators demonstrate strong commitment and reliability in using digital health systems. Supervisors stated that women are more responsive to feedback, and more proactive in taking corrective actions compared to men.

**Vaccinators' perspectives.** Findings showed that most vaccinators, including Health Extension Workers are females. Vaccinators mentioned that they did not observe or experience any gender focused intervention at their work environment by their facility or supervisors. Moreover, they had not encountered any supervision approaches that specifically considered gender context or needs. The majority of vaccinators reported having no preference for a supervisor's gender and had not faced any challenges related to their supervisors. One participant stated:

*"I haven't seen or experienced any activities focused on gender at work. Most of the time, our supervisors are male, and I haven't had any issues with them. I don't have a preference for a supervisor's gender." (P017, vaccinator, F)*

The findings from the vaccinators indicated that they use EIR and eCHIS for immunization service delivery and find them helpful. However, they noted that the implementation process is the same for all users, regardless of gender, and they have not observed any gender-focused approach.

A vaccinator described the idea as:

*"As for me, I believe the digital health tools are easy to implement, and we receive training and supervision. However, I haven't seen any gender-focused approach in the training, supervision, or any other practices; it all seems to follow the same approach for everyone." (P016, vaccinator, F)*

## Gender-intentional data management and decision-making

The study revealed that sex-disaggregated data is commonly collected in the immunization program, particularly through national surveys, to assess service utilization and coverage among male and females. Additionally, the health system incorporates approximately 13 gender-related indicators designed to monitor gender-sensitive practices across various health programs, including immunization and digital health. These indicators cover key areas such as the presence of a

gender and social affairs office within management structures, the integration of gender and social issues into programs, availability of gender-related policies and guidelines, gender-balanced participation in trainings and meetings, GBV prevention and response activities, accessibility for persons with disabilities, support services for female employees like childcare facilities, women's participation in leadership, and the implementation of affirmative actions for women and marginalized groups. A detailed list of these indicators is provided (Table 2).

Participants also highlighted that while survey tools often include sex-disaggregated questions, the involvement of women throughout the survey process from planning to field operations and data analysis remains limited. They noted that men predominantly lead these processes, with fewer women actively participating. Additionally, the demanding nature of data collection activities poses challenges that further discourage female participation in survey implementation, limiting their mobility, and raise safety concerns. A participant explained:

**Table 2. Gender focused indicators included in the health system Integrated Supportive Supervision (ISS) checklist, 2024 Ethiopia.**

| No | Themes | Indicators |
|----|--------|-----------|
| 1. | Organizational structure | Is there a designated office/unit for gender and social affairs implementation? Is it part of management? If the answer is no/not a member, please specify the reason. |
| 2. | Inclusion | Are gender, children, youth, and social issues integrated as an institution and program? If not, please specify the reason. |
| 3. | | Is there a designated budget for the gender and social affairs implementation office/unit? If not, please specify the reason. |
| 4. | | Are policies, strategies, and guidelines related to gender, women, persons with disabilities, workplace harassment prevention and response, and similar issues available? Are they recognized at different levels? |
| 5. | | Do the training sessions, meetings, and research activities conducted by the health bureau/office include the participation of women and youth? If not, please specify the reason? |
| 6. | Gender-Based Violence (GBV) Prevention and Response | Are awareness-raising activities conducted on gender-based violence (GBV) prevention and response? If not, please specify the reason. |
| 7. | | In institutional supportive supervision visits, is gender included as a component? If not, please specify the reason. |
| 8. | Accessibility and Inclusion for Persons with Disabilities | Is the institution accessible and accommodating for persons with disabilities? |
| 9. | | Are efforts being made to ensure that persons with disabilities and other socially marginalized groups receive services that meet their special needs (e.g., capacity-building training for professionals, sign language interpretation)? If not, please specify the reason. |
| 10. | Support for Women | Is there a childcare facility for female employees to keep and breastfeed their children? If not, please specify the reason? |
| 11. | | Are women participating in leadership and decision-making positions? |
| 12. | | Are affirmative action measures in place to support women and persons with disabilities in recruitment, promotion, professional development, and short-/long-term training opportunities? If not, please specify the reason. |
| 13. | | What activities have been implemented to empower women and promote them into decision-making roles? |

*"… Survey tools are comprehensive and include gender-related data. However, participation remains male-dominated as data collection is time-consuming, involves extensive travel, and poses higher risks for females" (P038, policy maker, F)*

However, a gap exists in the national health information system regarding sex-disaggregated immunisation data. Although sex is routinely captured at the point of service delivery using primary data collection tools such as immunization registers, DHIS2 immunisation modules do not aggregate or report this information due to the absence of a sex variable in the system. Consequently, sex-disaggregated facility-level data cannot be transferred into DHIS2, which limits the analysis and interpretation of immunisation indicators disaggregated by sex and constrains their use for monitoring, planning, and decision-making.

Despite ongoing training and supervision activities, sex-disaggregated records of trainees and supervisees are not maintained. At service delivery points, there is no system to record the sex of caregivers; immunization records typically capture only the mother's name, with no option to document information about fathers, partners, or other relatives who bring the child for vaccination. As a result, the current registration system cannot capture this important data when someone other than the mother accompanies the child.

### Enablers of gender-intentional inclusiveness Digital Health for Immunization

This assessment identified key enablers to improve gender inclusiveness in Digital Health Information (DHI), including national priority, user-friendly digital tools, capacity-building training, gender-sensitive policies and supportive structures, empowerment and leadership opportunities for women, and partner engagement and support. Most of the participants stated national priority and attention to enhance women's engagement and participation in different programs and developing different implementation manuals as key enablers to improve gender inclusiveness in DHI. The participants also highlighted that the Ethiopian government has prioritized digital health transformation, ensuring that the health system's digital interventions are accessible to healthcare workers, regardless of gender. As one expert noted:

*"The government is strongly committed to improving the health system through digital transformation, ensuring that everyone, regardless of gender, has access to and benefits from these systems."(P029, policy maker, M)*

In addition, the involvement of various partners has been discussed as another key enabler for the implementation of gender inclusive digital health systems It was noted that partners provide technical and financial support to help ensure that systems are gender inclusive and accessible to all. Other participants highlighted that the empowerment of women in the health sector is another enabler for gender inclusiveness in digital health. Participants mentioned there has been increasing recognition of the importance of empowering women to take on leadership and technical roles within the healthcare system. One expert pointed out:

*"Recently, the health system has increasingly acknowledged and recognized the importance of empowering women to take on various positions and leadership roles." (P037, gender expert, F).*

Some respondents highlighted that capacity-building training programs focused on women to strengthen their leadership and management skills as an opportunity that would further support the implementation of gender-inclusive programs. They noted that affirmative action measures are in place within the health system. These measures are practiced in recruitment processes for immunization and digital health positions, as well as professional development opportunities for staff. An expert said that:

*"We have developed women empowerment guidelines and various training programs, including leadership incubation program that targets a 60% female participation rate. In terms of education and training, the constitution and civil service proclamation provide an additional 5% consideration for women..." (P042, gender expert, F)*

### Barriers to gender-intentional Digital Health for Immunization

The assessment also identified key barriers that require intervention to enhance gender inclusiveness, including cultural and societal perceptions, lack of awareness of the importance of gender factors, limited advocacy, weak leadership support and engagement, low resource and partner involvement, weak monitoring and evaluation (M&E) systems for gender-related activities, and a shortage of skilled human resources in gender-related areas.

Many experts and stakeholders in the health sector acknowledge that gender is not well understood, and existing gender advocacy efforts face challenges in achieving their intended impact. One respondent remarked,

*"Unfortunately, I don't think we have had awareness or deliberate efforts to address gender considerations in immunization-related planning and implementation processes. We are simply following the usual approach without incorporating specific gender-focused strategies." (P030, policy maker, M).*

Most participants noted that cultural and societal perceptions create significant barriers to gender inclusiveness in digital health for immunization. Traditional gender roles often limit women's participation in leadership and supervisory positions and also in adopting digital health intervention, as they are expected to prioritize household responsibilities over career advancement. As one respondent noted,

*"...the roles and responsibilities at her household tend to prevent women from traveling to different places to take trainings and hold meetings that will help them develop their career." (P039, gender expert, F).*

Despite the inclusion of gender considerations in some health strategies, participants noted a lack of robust advocacy to ensure their meaningful integration into digital health for immunization programs. One participant commented,

*"Gender is mentioned in some health strategies, but there is no strong advocacy to ensure its meaningful integration in digital immunization programs." (P040, gender expert, F)*

Limited human resources and funding hinder the implementation of gender-inclusiveness. This is compounded by a lack of prioritization of gender issues within organizations. A participant mentioned that,

*"Furthermore, the actual practice of implementing gender-responsive interventions at is hindered by low awareness, weak commitment, and resource constraints." (P041, gender expert, F)*

## Discussion

This study explored gender inclusiveness in Ethiopia's DHI for immunization system and identified gaps and progress across multiple domains. The study revealed a marked gender imbalance in the DHI landscape for immunization; women are highly represented as vaccinators and caregivers but remain underrepresented in leadership, supervision, and digital system development roles. The gender analysis further revealed that while strategic documents acknowledge gender issues and gender units exist in the health system, the practical implementation is weak across DHI for immunization programs. Most stakeholders demonstrated limited understanding of gender beyond male-female distinctions, with

little or no prior exposure to gender related trainings. Role assignment continues to rely on availability and competency without deliberate efforts to consider gender. Despite widespread collection of sex-disaggregated data, gaps remain in gender-intentional data management and decision making, particularly in DHIS2 and supervision systems. Enablers such as national prioritization, gender sensitive policies, and empowerment programs offer foundation for progress. Yet, barriers persist, including cultural and societal perceptions, limited advocacy, lack of awareness, weak leadership support and engagement, low resource and partner involvement, and weak monitoring and evaluation. These barriers continue to impede meaningful gender inclusiveness in DHI for immunization.

The study demonstrated that women's involvement in digital health interventions development in Ethiopia is minimal. These findings resonate with global trends in digital health and broader health systems where women are essential to health service delivery but often excluded from decision making and digital innovations [20]. The vertical occupational gender segregation observed in Ethiopia's DHI for immunization is consistent with patterns documented in broader high-tech sectors, where women are frequently underrepresented in leadership and technical roles while occupying support and lower-status positions [21]. According to the World Health Organization (WHO), women constitute roughly 70% of the health workforce, delivering much of the community health service, and women in health leadership are roughly at 25% [22]. Similar trends have been reported in other low-and-middle income countries (LMICs) where women dominate operational roles but lack representation in digital system governance and policy making [23,24]. The situation is further compounded by the societal barriers limiting women's opportunities to participate in high-level decision-making and digital health innovation, despite being the primary users of digital tools like EIR or eCHIS, including restrictive gender norms, limited mobility, digital literacy gaps, and competing household responsibilities [25].

Evidence shows that digital tools developed without meaningful female participation frequently overlook gender specific barriers, such as disparities in mobile access, digital literacy, burden of household responsibilities and cultural norms that limit meaningful participation and use of digital approaches [2,26]. Although the high representation of women in frontline services is a deliberate alignment with the predominantly female caregivers they serve, the significant disparity, where are women are overwhelmingly the end-users and frontline implementers of DHI, yet are minimally involved in its design, leadership, and policy making, creates a fundamental disconnect that compromises the effectiveness and user centricity of DHI systems. The inclusion of gender training as part of the design and implementation of digitally-enabled microplanning for immunization in the Democratic Republic of Congo showed promising results, including more support for women to participate as vaccinators and in the microplanning process as well as improved community awareness and mobilization [27]. The gender digital divide in mobile phone ownership also has a critical implication for immunization and primary health care outcomes. Evidence highlights that women's mobile phone access is significantly associated with their ability to receive health information and engage with digital tools and correlates with improved immunization rates and health service utilization [28]. In Ethiopia, only 52% of women own mobile phones, hindering equitable access to digital health benefits [16]. Addressing this gap through targeted digital inclusion strategies is essential for achieving gender-equitable and effective digital health systems.

The gender dynamic analysis further revealed that digital health and immunization strategic documents recognize gender gaps and outline proposed solutions. In addition, gender-specific departments are embedded within the health system from the Ministry of Health to woreda health offices. Ethiopia's efforts to integrate gender into its immunization strategies reflect growing recognition of gender gaps, similar to initiatives in other countries like Rwanda [11], Kenya [10] and Bangladesh [29]. However, while gender issues are prioritized in the health system, the practical implementation is weak across DHI for immunization programmes, highlighting that Ethiopia's digital health initiatives are still at the gender-sensitive stage, at best. This implementation failure is not unique to Ethiopia; a multicounty review noted gender policies often exist on paper without corresponding enforcements, monitoring or budgeting mechanisms [2]. Gender units also often lack authority or resources, as a result, even if women empowerment initiatives like affirmative action measures (5% recruitment bonus for female health workers) exist, these initiatives have not been institutionalized and sustained

across the digital health and immunization programs. This results in an inability to create significant change in gender inclusiveness and remain insufficient to address deep-rooted inequalities in opportunity and access. To move beyond the gender-sensitive stage, it is better to focus on strengthening policy frameworks, promoting cultural change to challenge norms, and expanding capacity-building efforts for women to take on leadership roles.

Another consistent theme of the findings was the limited awareness and understanding of gender beyond male-female distinctions among stakeholders in DHI for immunizations. Few stakeholders demonstrated an understanding of gender as a social construct shaped by norms, power relations, and structural inequalities. This limited understanding was attributed to the lack of prior exposure to gender-related training across the health system. The findings align with other studies in Kenya and Bangladesh, where although gender equality is a national priority, the actual understanding of gender concepts among health professionals and implementers remained weak [10,30]. Similarly in Nigeria, it was documented how program managers and data officers working on immunization had not received any gender training and were unable to interpret data, missing insights into inequities [31]. As a result, programs are inconsistently gender-integrated with lack of clarity on how gender influences opportunities, decision-making, and participation in health programs. A critical step to gender intentional DHI for immunization is to understand gender inequalities and the power relations that underpin them, thus the limited understanding of gender among stakeholders in Ethiopia leads to several practical blind spots. For instance, immunization records do not capture the sex of caregivers, reinforcing the assumption that only mothers bring children for vaccination. Without strengthening gender literacy across the health system, including developers and decision makers, Ethiopia's DHI efforts risk reinforcing rather than reducing inequality.

The study also found that role assignment within Ethiopia's DHI for immunization system is based on candidate availability and competency, yet this approach might unintentionally reinforce structural gender inequities. Women, who face greater constraints due to household responsibilities, safety concerns, and limited mobility, are underrepresented in supervisory, leadership, and digital technical roles, even though they make up the majority of frontline vaccinators [32]. Evidence showed that gender neutral allocation often resulted in a male-dominated leadership due to gendered barriers that were left unaddressed [25]. This oversight can perpetuate inequalities and restrict opportunities for women in essential areas of immunization and digital health, ultimately undermining the effectiveness of these programs.

Although sex-aggregated data are routinely collected through national surveys and supervision checklists in Ethiopia, critical gaps remain in gender-intentional use of data for program management and decision-making. Specifically, the DHIS2 platform lacks a dedicated sex variable for immunization, making it impossible to disaggregate key indicators by sex. Likewise, records do not capture the sex of caregivers who bring children for vaccination, nor do training and supervision records consistently track participants' gender. The lack of gender-disaggregated data may be a function of the overall maturity of the health information system, it is anticipated that this limitation may be addressed as the country matures and adopts more longitudinal records. These limitations prevent program managers from identifying gender disparities in service delivery, capacity building, and care giver engagement, thereby undermining efforts to design responsive interventions. A similar challenge has been observed in Kenya, where despite the inclusion of gender-responsive language in national immunization policies, operational data systems did not adequately capture gender-related variables [10]. Evidence underscores the importance of integrated gender disaggregated data into digital systems and linking to decision making [20]. As decision makers need gender disaggregate data to inform policies that seek to close any gender gap in digital health, Ethiopia would benefit from not only improving the technical structure of DHIS2 to include gender variables, but also build the analytical and institutional capacity to routinely use gender data in planning, resource allocation, and accountability.

Findings highlight both promising developments and persistent challenges in gender inclusiveness within Ethiopia's DHI for immunization. The study identified several important enablers that could support Ethiopia's transition toward a more gender-intentional DHI system for immunization such as national prioritization of gender provides a strong policy foundation. The existence of gender-sensitive policies and dedicated gender units within the health system also reflects

growing institutional commitment to addressing gender disparities. These enabling factors align with other countries such as Bangladesh and Kenya, where strategic policy frameworks and gender units laid similar foundation for promising equity in digital health systems [10,29]. Additionally the presence of empowerment programs and targeted initiatives designed to enhance the capacity and participation of women provides further momentum for advancing inclusivity in DHI for immunization systems.

Despite these enablers, several entrenched barriers continue to limit meaningful gender integration. Cultural and societal norms remain significant barriers, often reinforcing traditional roles that marginalize women from leadership, supervisory, and technical positions within the DHI system. The study also revealed that gender is still narrowly understood by many stakeholders, with limited exposure to gender related training and advocacy efforts. This limited conceptual understanding contributes to lack of intentionality in role assignments and programmes design. Another major barrier is the weak support and engagement from leadership in translating gender-sensitive policies into actionable strategies within the DHI system. Resource constraints further exacerbate this issue, as gender inclusive initiatives often suffer from insufficient funding and minimal involvement from key partners. The absence of robust monitoring and evaluation mechanisms specifically focused on gender dimensions, within the DHI hinders accountability and the ability to track progress toward gender equity goals. Taken together, these findings highlight critical gaps and opportunities for advancing gender equity in Ethiopia's DHI for immunization systems.

A major strength of this study lies in the use of diverse qualitative methods including document reviews, stakeholder analysis matrix, workshops and interview, which enabled a comprehensive and multi-level perspective on gender dynamic. Notably, as the first study to examine gender inclusiveness within DHI for immunization in Ethiopia, it offers critical insights that can guide future policy and practices. However, the study has limitation. Although it included diverse regions, it did not capture majorly pastoralist areas, where digital access and gender norms may differ, due to limited availability of digital health interventions.

## Conclusion

This study highlights persistent gender imbalance in Ethiopia's DHI for immunization, despite the existence of gender policies and structures. Women dominate caregiving and frontline vaccination roles but remain underrepresented in supervisory, leadership, and digital health system development positions. Although national strategies, guidelines, and institutional structures reflect a growing recognition of gender issues, this commitment has not yet been effectively translated into practice. Gender awareness and understanding among stakeholders remains limited, role assignment within the health system continues to be driven by availability and competency without acknowledging gender related barriers. While sex-disaggregated data are collected through surveys and gender indicators are included in supervision checklist, critical limitation exist in systems like DHIS2, which lacks dedicated gender variables to inform decisions. Enablers such as strong political will, gender-sensitive policies, and women's empowerment initiatives provide a promising foundation. However, these are undermined by persistent barriers, including cultural and social norms, limited advocacy and awareness, weak leadership engagement, insufficient funding, and inadequate monitoring and evaluation systems.

Despite some progress, the overall impact of gender inclusiveness in immunization and digital health programs in Ethiopia remains limited, highlighting the urgent need to close the gap between policy and practice. To advance gender equity and move forward with a gender-transformative DHI system in Ethiopia, a systematic and sustained shift is required. This includes institutionalize gender trainings to raise gender awareness and capacity across all levels of the DHI system. The government needs to focus on turning the existing policies into real actions by setting up monitoring systems to ensure they are effectively integrated into immunization and digital health systems, with accountability for their impact. In addition, Gender units are essential for promoting gender inclusivity but often lack sufficient funding and staff. Strengthening these units in Ministry and Regional Health Bureau by providing enough resources, proper staffing, and specific training will help ensure the successful implementation of gender-focused programs. Addressing structural and societal barriers by tackling

the root causes of the issue, starting with early childhood development and the primary education process, extending to household and community levels.

Creating mentorship and leadership programs aimed at women is also essential to balance gender representation and empower women to take leadership and decision-making roles in health systems. Integrating gender-disaggregated data into decision-making and robust monitoring and evaluation systems, along with strong stakeholder engagement, will also be key to ensuring digital health interventions are equitable and drive inclusive progress toward improved health system and immunization outcomes. To fully realize the potential of digital health in Ethiopia's immunization program, it is essential to move beyond viewing women as passive users of digital approaches and begin to address the systematic disparities by ensuring their meaningful involvement in leadership, design, and decision making.

## Supporting information

**S1 Table. Gender integration in immunization and digital health policies in Ethiopia.**
(DOCX)

**S2 Table. Illustrative participant quotes supporting key themes.**
(DOCX)

## Acknowledgments

We acknowledge the institutional support received from GDI Solutions and health.enabled, which enabled the smooth execution of this research. We would like to acknowledge the participants, including the health system stakeholders, vaccinators, supervisors, policymakers, and digital health experts whose invaluable insights and experiences made this research possible.

## Author contributions

**Conceptualization:** Getasew Amare, Betelhem Abebe Andargie, Patricia Mechael, Xueli Qiu, Michelle R. Kaufman, Berhanu Fikadie Endehabtu, Erica Layer, Sarah Cunard Chaney, Binyam Tilahun.

**Data curation:** Getasew Amare, Betelhem Abebe Andargie, Berhanu Fikadie Endehabtu, Kassahun Alemu.

**Formal analysis:** Getasew Amare, Betelhem Abebe Andargie, Patricia Mechael, Berhanu Fikadie Endehabtu, Binyam Tilahun.

**Funding acquisition:** Patricia Mechael, Erica Layer.

**Investigation:** Getasew Amare, Betelhem Abebe Andargie, Berhanu Fikadie Endehabtu, Kassahun Alemu.

**Methodology:** Getasew Amare, Betelhem Abebe Andargie, Patricia Mechael, Xueli Qiu, Michelle R. Kaufman, Berhanu Fikadie Endehabtu, Erica Layer.

**Project administration:** Betelhem Abebe Andargie, Erica Layer, Sarah Cunard Chaney.

**Supervision:** Getasew Amare, Patricia Mechael, Michelle R. Kaufman, Erica Layer, Binyam Tilahun.

**Validation:** Getasew Amare, Betelhem Abebe Andargie, Patricia Mechael, Xueli Qiu, Berhanu Fikadie Endehabtu, Erica Layer, Binyam Tilahun.

**Writing – original draft:** Getasew Amare, Betelhem Abebe Andargie, Patricia Mechael, Xueli Qiu, Michelle R. Kaufman, Berhanu Fikadie Endehabtu, Kassahun Alemu, Erica Layer, Sarah Cunard Chaney, Binyam Tilahun.

**Writing – review & editing:** Getasew Amare, Betelhem Abebe Andargie, Patricia Mechael, Xueli Qiu, Michelle R. Kaufman, Berhanu Fikadie Endehabtu, Kassahun Alemu, Erica Layer, Sarah Cunard Chaney, Binyam Tilahun.

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
