## [Decision Letter · Decision Letter 0]

4 Jan 2026

PGPH-D-25-03223

Exploring Gender-Intentional Implementation of Digital Health Information for Immunization in Ethiopia

Dear Mechael,

Thank you for submitting your manuscript to PLOS Global Public Health. After careful consideration, we feel that it has merit but does not fully meet PLOS Global Public Health’s publication criteria as it currently stands. Therefore, we invite you to submit a revised version of the manuscript that addresses the points raised during the review process.

We look forward to receiving your revised manuscript.

Kind regards,

Collins Otieno Asweto, PhD

Academic Editor

Journal Requirements:

1. Please provide a detailed online Financial Disclosure statement. This is published with the article. It must therefore be completed in full sentences and contain the exact wording you wish to be published.

a) State the initials, alongside each funding source, of each author to receive each grant. For example: “This work was supported by the National Institutes of Health (####### to AM; ###### to CJ) and the National Science Foundation (###### to AM).”

For more information, please go to our submission guidelines:

https://journals.plos.org/globalpublichealth/s/submission-guidelines#loc-financial-disclosure-statement

2. Please ensure that the funders and grant numbers match between the Financial Disclosure field and the Funding Information tab in your submission form. Note that the funders must be provided in the same order in both places as well.

3. Please update your online Competing Interests statement. If you have no competing interests to declare, please state: “The authors have declared that no competing interests exist.”

4. We note that your Data Availability Statement is currently as follows: “All data generated or analyzed during this study are included in this article and its supplementary information files”

Please confirm at this time whether or not your submission contains all raw data required to replicate the results of your study. Authors must share the “minimal data set” for their submission. PLOS defines the minimal data set to consist of the data required to replicate all study findings reported in the article, as well as related metadata and methods (https://journals.plos.org/globalpublichealth/s/data-availability#loc-minimal-data-set-definition).

If your submission does not contain these data, please either upload them as Supporting Information files or deposit them to a stable, public repository and provide us with the relevant URLs, DOIs, or accession numbers. For a list of recommended repositories, please see https://journals.plos.org/globalpublichealth/s/recommended-repositories.

Additional Editor Comments (if provided):

Reviewers' comments:

Reviewer's Responses to Questions

**Comments to the Author**

1. Does this manuscript meet PLOS Global Public Health’s publication criteria? Is the manuscript technically sound, and do the data support the conclusions? The manuscript must describe methodologically and ethically rigorous research with conclusions that are appropriately drawn based on the data presented.? Is the manuscript technically sound, and do the data support the conclusions? The manuscript must describe methodologically and ethically rigorous research with conclusions that are appropriately drawn based on the data presented.

Reviewer #1: Yes

Reviewer #2: Yes

Reviewer #3: Yes

2. Has the statistical analysis been performed appropriately and rigorously?

Reviewer #1: N/A

Reviewer #2: Yes

Reviewer #3: N/A

3. Have the authors made all data underlying the findings in their manuscript fully available (please refer to the Data Availability Statement at the start of the manuscript PDF file)?

The PLOS Data policy requires authors to make all data underlying the findings described in their manuscript fully available without restriction, with rare exception. The data should be provided as part of the manuscript or its supporting information, or deposited to a public repository. For example, in addition to summary statistics, the data points behind means, medians and variance measures should be available. If there are restrictions on publicly sharing data—e.g. participant privacy or use of data from a third party—those must be specified.requires authors to make all data underlying the findings described in their manuscript fully available without restriction, with rare exception. The data should be provided as part of the manuscript or its supporting information, or deposited to a public repository. For example, in addition to summary statistics, the data points behind means, medians and variance measures should be available. If there are restrictions on publicly sharing data—e.g. participant privacy or use of data from a third party—those must be specified.

Reviewer #1: Yes

Reviewer #2: Yes

Reviewer #3: Yes

4. Is the manuscript presented in an intelligible fashion and written in standard English?

Reviewer #1: Yes

Reviewer #2: Yes

Reviewer #3: Yes

Reviewer #1: Manuscript Number: PGPH-D-25-03223

Exploring Gender-Intentional Implementation of Digital Health Information for

Immunization in Ethiopia

General Comments:

The study titled “Exploring Gender-Intentional Implementation of Digital Health Information for

Immunization in Ethiopia” is of great interest.

In the contemporary era of rapid technological advancement and shifting global demographics, women constitute a substantial proportion of the population worldwide and continue to serve as the primary caregivers and care providers. In the context of immunization, mothers traditionally bear the greatest responsibility for child healthcare decision-making and service utilization. However, the inclusion of fathers in immunization-related processes remains critically important to promote shared responsibility and improved health outcomes.

Therefore, the integration and implementation of a gender-intentional framework is an essential priority for immunization research and practice. The insights and lessons generated from this study will not only strengthen local immunization strategies but will also be applicable to other resource-constrained settings with similar sociocultural and health system contexts.

The paper is well-written scientifically.

I am recommending publishing this manuscript with minor changes.

Methods:

The process of reaching and recruiting participants could be further elaborated for future researchers.

Please clarify the methodological approaches used to minimize reporting bias and to account for the authors’ positionality and potential biases

Reviewer #2: Thank you very much for the opportunity to review this manuscript. This manuscript is well written and has an important topic for public health issues, especially immunization program. I have some comments to improve the quality of the manuscript.

Abstract

Authors stated DHIS2, please don’t use abbreviations in the first word

Methods

Authors should describe in detail the key stakeholders involved in this study

Authors should add information how many participants in this study

Authors should explain detail each method conducted in this study such as a participatory design workshop, interviews, and stakeholders mapping. Who is participating in each method? Please explain them detail

Results

Authors explained that they involved eight documents to conduct this study. Authors should mention them in detail

In the subheading gender dynamics in digital health information for immunization, authors stated that Women’s involvement in digital health interventions development is minimal. Authors should add an evidence-based to support this statement.

Authors stated that the gap is observed in DHIS2 system, which lacks a dedicated sex variable for the immunization program. As a result, immunization indicators cannot be analyzed and interpreted from a sex or gender perspective, even when source documents or registers capture sex disaggregated data. Could you clarify this statement? And please add the evidence to support this statement.

Reviewer #3: The overall study is highly significant considering that there is limited evidence around this. The author has made an excellent effort to address this issue.

1: The Introduction section can be elaborated more with detailed literature review & background.

2: Line 46-54 can be readjusted and place it after line 55-61.

3: It would be good if the author mentions how many participants were approached and how many of them refused to take part in the study.

4: Inclusion of COREQ checklist is highly recommendable.

**Do you want your identity to be public for this peer review?** For information about this choice, including consent withdrawal, please see our Privacy Policy..

Reviewer #1: No

Reviewer #2: No

Reviewer #3: No

---

## [Decision Letter · Decision Letter 1]

5 Mar 2026

Exploring Gender-Intentional Implementation of Digital Health Information for Immunization in Ethiopia

PGPH-D-25-03223R1

Dear Patricia,

We are pleased to inform you that your manuscript 'Exploring Gender-Intentional Implementation of Digital Health Information for Immunization in Ethiopia' has been provisionally accepted for publication in PLOS Global Public Health.

Best regards,

Collins Otieno Asweto, PhD

Academic Editor

Reviewer's Responses to Questions

**Comments to the Author**

Reviewer #1: All comments have been addressed

Reviewer #2: All comments have been addressed

Reviewer #3: All comments have been addressed

publication criteria? Is the manuscript technically sound, and do the data support the conclusions? The manuscript must describe methodologically and ethically rigorous research with conclusions that are appropriately drawn based on the data presented.? Is the manuscript technically sound, and do the data support the conclusions? The manuscript must describe methodologically and ethically rigorous research with conclusions that are appropriately drawn based on the data presented.

Reviewer #1: Yes

Reviewer #2: Yes

Reviewer #3: Yes

3. Has the statistical analysis been performed appropriately and rigorously?

Reviewer #1: N/A

Reviewer #2: Yes

Reviewer #3: N/A

4. Have the authors made all data underlying the findings in their manuscript fully available (please refer to the Data Availability Statement at the start of the manuscript PDF file)?

The PLOS Data policy requires authors to make all data underlying the findings described in their manuscript fully available without restriction, with rare exception. The data should be provided as part of the manuscript or its supporting information, or deposited to a public repository. For example, in addition to summary statistics, the data points behind means, medians and variance measures should be available. If there are restrictions on publicly sharing data—e.g. participant privacy or use of data from a third party—those must be specified.requires authors to make all data underlying the findings described in their manuscript fully available without restriction, with rare exception. The data should be provided as part of the manuscript or its supporting information, or deposited to a public repository. For example, in addition to summary statistics, the data points behind means, medians and variance measures should be available. If there are restrictions on publicly sharing data—e.g. participant privacy or use of data from a third party—those must be specified.

Reviewer #1: Yes

Reviewer #2: Yes

Reviewer #3: Yes

5. Is the manuscript presented in an intelligible fashion and written in standard English?

Reviewer #1: Yes

Reviewer #2: Yes

Reviewer #3: Yes

Reviewer #1: The authors have addressed all comments. However, it would be appreciated if the DOI/URL for all references could be added.

Reviewer #2: Authors already addressed reviewer comments

Reviewer #3: The author has responded the comments.

**Do you want your identity to be public for this peer review?** For information about this choice, including consent withdrawal, please see our Privacy Policy..

Reviewer #1: No

Reviewer #2: No

Reviewer #3: No
